# Allogeneic Hematopoietic Stem Cell Transplantation as a Platform to Treat Chemorefractory Acute Myeloid Leukemia in Adult Patients

**DOI:** 10.3390/cancers17203285

**Published:** 2025-10-10

**Authors:** Daniel Alzetta, Irene M. Cavattoni, Federico Mosna

**Affiliations:** Hematology and BMTU, Hospital of Bolzano (SABES–ASDAA), Teaching Hospital of Paracelsus Medical University (PMU), Via Lorenz Bohler 5, 39100 Bolzano, BZ, Italy; daniel.alzetta@sabes.it (D.A.);

**Keywords:** acute myeloid leukemia, chemoresistance, hematopoietic stem cell transplantation, intensified conditioning, sequential conditioning, immunotherapy, Graft-versus-leukemia, GvL, HLA-match, donor selection, TP53-mutated AML, DLI

## Abstract

**Simple Summary:**

Results in the treatment of adult patients affected by chemorefractory acute myeloid leukemia are poor, with median survival around 2–4 months for patients ineligible for allogeneic hematopoietic stem cell transplantation and long-term survival in the order of 20–30% for those undergoing transplantation. It is widely debated how to make transplantation more effective. Increasing the dose-intensity by sequencing high-dose chemotherapy and reduced-intensity conditioning with transplantation performed during aplasia has been tested as a potential strategy, as well as other approaches centered on the optimization of alloreactivity of donor T- and NK-cells against host leukemia. This can be achieved by donor selection, accelerated withdrawal of immunosuppression, and the administration of donor lymphocyte infusion as prophylaxis or pre-emptive therapy of leukemia relapse. In this review, we will critically summarize the main tested strategies and address the yet unresolved issues.

**Abstract:**

Adult patients affected by acute myeloid leukemia who fail to achieve remission after two cycles of intensive chemotherapy based on a combination of anthracyclines and cytarabine are considered chemorefractory and are unlikely to benefit from further induction attempts. Characterized by a poor prognosis, they may still benefit from allogeneic hematopoietic stem cell transplantation, even if long-term survival rarely exceeds 20–30%. Still, the use of sequential high-dose chemotherapy followed by reduced-intensity conditioning, with transplantation performed during aplasia, and the optimization of the alloreactivity of donor leukocytes against leukemia (i.e., the graft-versus-leukemia effect) may ameliorate these results. Optimization of alloreactivity against leukemic cells can be achieved by proper donor selection, by the early withdrawal of immunosuppressive therapy, by post-transplant administration of donor lymphocyte infusions as prophylaxis of leukemia relapse, and by several other maintenance and preemptive therapies. Far from being the final stage of consolidation therapy, allogeneic hematopoietic stem cell transplantation is now considered as the moment when a unique immunological platform can be established in these patients, to be used for additional post-transplant measures. In this study we will critically review the different pre- and post-transplant strategies used in clinical trials to improve long-term survival in adult patients transplanted with chemorefractory leukemia.

## 1. Introduction

Despite being generally considered a chemosensitive disease, acute myeloid leukemia (AML) proves resistant to first induction and rescue treatments in up to 30% of genetically defined intermediate-risk and 50% of high-risk adult patients [1,2]. In fact, the patients failing to achieve complete remission (i.e., <5% leukemic blasts in the bone marrow, in the absence of extramedullary disease, with or without a complete hematological recovery) after two courses of intensive induction attempts are defined according to the most recent guidelines [2,3] as chemorefractory and characterized by severe prognosis [4]. How to best treat them is still an unresolved issue.

Allogeneic HSCT has proven to be a valid potential therapeutic option in extensive clinical practice, as well as large meta-analyses [5,6], even though long-term results are poor even with this approach, as 5-year survival rates commonly range from 20 to 30% [1,2,7,8,9,10,11,12,13,14,15,16,17,18,19,20,21]. This is thought to be the consequence of its unique properties, such as (1) the possibility to provide maximal chemotherapy and/or total body irradiation (TBI) against leukemic cells, thus potentially overcoming intrinsic resistance; (2) the transfer of the donor’s immunological system and subsequent activity of donor T- and NK-cells against residual AML cells, a phenomenon known as the “Graft-versus-Leukemia” (GvL) effect; and (3) the reconstitution of a healthy allogeneic bone marrow environment to potentially serve as an immunological platform to exploit additional therapies.

In this review we will critically address several strategies tested in recent years to overcome resistance and improve long-term results in adult chemorefractory AML patients undergoing allogeneic HSCT. We will argue how, far from its original concept of final consolidation treatment, allogeneic HSCT is now considered as the moment when a unique immunological platform can be established, to be used for additional post-transplant immunotherapeutic measures.

## 2. Methods

To address the role of allogeneic HSCT in the treatment of chemorefractory AML patients, we performed a systematic review of the literature by searching the main articles published on this topic in the last 25 years by means of several keywords in PubMed and other journal-specific search engines. These included the following: allogeneic hematopoietic stem cell transplantation, bone marrow transplantation, refractory acute myeloid leukemia, myeloablative conditioning regimen, sequential conditioning regimen, rescue treatment, umbilical cord blood, TP53-mutated, p53, 17p-deletion, and therapy of persistent acute myeloid leukemia. All major prospective and retrospective studies published in peer-reviewed journals by international cooperative groups, as well as all major registry studies, were summarized and critically included in the analysis. We systematically excluded research published only in abstract form, single case reports, and small (i.e., <10 patients) clinical experiences reported by single centers. The aim of this review is to suggest rational strategies for the improvement of the outcome of chemorefractory adult AML patients. Conclusions were drawn by the authors based on evidence, common clinical practice, and personal experience and represent only the authors’ opinion, as stated in the present article.

## 3. Allogeneic HSCT Performed After Conventional Conditioning

Several studies have described the outcome after allogeneic HSCT of patients with active AML: overall, results are significantly worse than what is achievable in patients transplanted in remission, with long-term survival ranging between 20% and 30% in most published series. This is due to an increased and earlier relapse incidence (RI) after transplantation, as well as to an overall increased infectious risk and toxicity, leading to higher non-relapse mortality (NRM) [7,8,9,10,11,12,13,14,15,16,17,18,19,20,21]. Table 1 summarizes the main studies published in the literature on patients transplanted with active AML after conventional conditioning.

Considering the available evidence, this consists primarily of large registry studies from national and international cooperative groups and a minority of prospective multicenter clinical trials; as such, in most cases results are derived from a variety of conditioning treatments, broadly divided into myeloablative (MAC), reduced intensity conditioning (RIC), and non-myeloablative conditioning (NMA), in patients often transplanted at different stages of disease (primary refractory to 1 or 2 attempts by intensive chemotherapy, i.e., “Primary induction failure” (PIF), vs. relapsed patients, either undergoing direct transplantation or after failure of a rescue attempt).

Furthermore, data are gathered over a very long period, thus introducing the potential biases of most patients being transplanted after MAC conditioning rather than RIC or NMA, usually at an earlier age than in more recent studies, and prevalently from either matched related donors (MRD) or unrelated donors (MUD) rather than haploidentical donors and umbilical cord blood units (UCB).

Here we summarize the most relevant results as follows:The French cooperative group “*Société Francaise de Greffe de Moelle*” (SFGM) reported their experience on 379 patients, 230 of whom were transplanted with active AML. Long-term survival was highly dependent on the status at HSCT, ranging from 32% in the case of patients transplanted in remission to 9% for those not undergoing rescue treatment before HSCT to 11% for patients refractory to rescue [7]. The authors found age less than 15 years, longer first complete remission (CR) duration, use of MRD instead of MUD, and the presence of a male donor to be associated with a better overall survival (OS).This is similar to what was reported by the Center for International Blood and Marrow Transplant Research (CIBMTR) on 2255 patients transplanted between 1995 and 2004 after MAC [8]. Here, the authors reported a 3-year OS of 19% at a median follow-up of 61 months. Despite this dismal outcome, they still could ascertain some factors associated with a slightly better prognosis, namely: (1) the length of initial CR (>6 months), (2) the absence of circulating blasts, (3) a better performance status (Karnofsky score ≥ 90%), (4) a favorable or intermediate karyotype, and (5) HSCT from MRD. The minority of patients undergoing HSCT with all these favorable features (i.e., 13% of the whole series) experienced a 3-year OS of 42%, opposite to that of the most adverse group, which made up the majority of the series and experienced a 3-year OS of only 6% [8].Similarly, Craddock and coll. reported a 5-year OS of 22% from 168 patients affected by primary refractory AML and transplanted either from MRD or MUD between the years 1994 and 2006 by means of mostly (*n* = 132) a standard MAC (TBI-based in 83 of them). Fewer than three courses of induction attempts, a lower percentage of bone marrow (BM) involvement, and patient CMV seropositivity were associated with a better OS: in fact, the group with the most favorable combination of these prognostic features showed a striking 5-year OS of 44% (+/− 11%). This is remarkably similar to what was reported in the study by the CIBMTR [9].Weisdorf and coll. analyzed 4682 patients transplanted either in second complete remission (CR2) (*n* = 1986), primary induction failure (PIF) (*n* = 1440), or after failure of one or more rescue treatments (*n* = 1256). Patients achieving CR2 more likely had de novo AML, better PS, and longer first remission (CR1) duration. Conversely, adverse cytogenetics were more common in patients experiencing PIF. 5-year OS adjusted for performance status (PS), cytogenetic risk, and donor type was 39% vs. 21% vs. 18%, respectively, with patients transplanted with PIF achieving a slightly better OS at 5 years than patients transplanted after refractory relapse (21% vs. 18%, respectively) [14].Poiani and coll. reported data from 2089 patients transplanted between 2000 and 2017 with active disease defined as either refractory (*n* = 972) or relapsed AML (*n* = 1117, with or without a reinduction attempt). This series was divided according to the cytogenetic risk stratification used by the British cooperative group Medical Research Council (MRC). Unsurprisingly, intermediate- and high-risk patients were more common (*n* = 1283 + *n* = 652 vs. *n* = 154, respectively) and experienced worse LFS (34% vs. 27% and 18%, *p* < 0.01), lower CR rate at day 100 after HSCT (79% vs. 69% and 61%, *p* < 0.01), and higher relapse rate (42% vs. 51% and 61%) than favorable-risk patients. No difference in survival could be observed when stratifying according to conditioning intensity [15].In a large registry analysis by the Acute Leukemia Working Party (ALWP) of the European Society for Blood and Marrow Transplantation (EBMT), Nagler and coll. reported an overall 2-year LFS of around 25% after MAC/RIC/FLAMSA-RIC and demonstrated a trend towards slightly better results in more recent years. This may be likely the outcome of the significant progress in HLA matching (i.e., genetic vs. serologic) and in the supportive care and infection control practices achieved in the last two decades [16].Yanada and coll. described the data of 6927 adults affected by AML and transplanted between 2001 and 2020: 5-year OS, RI, and NRM were 23%, 53%, and 27%, respectively. At multivariate analysis, adverse cytogenetics, the percentage of circulating blasts, and the year of transplantation affected RI, while age, sex, and performance status affected NRM. Relapsed refractory disease was prognostically worse than PIF with regard to OS (24%, 95% C.I. 22–26 vs. 23%, 95% C.I. 21–24, *p* = NS, respectively). In the relapsed group, shorter duration of CR1 correlated with increased relapse risk and mortality (with 5-year OS ranging from 38% for patients with CR1 lasting > 24 months to 16% for those with CR1 < 6 months, *p* < 0.001), while in the PIF group the same was true using time from diagnosis to HSCT as covariate (from 5-year OS 27% for those with duration < 3 months vs. 21% for those ≥6 months, *p* < 0.001). Neither donor type nor the intensity of conditioning had a significant effect on OS [18].Jiang and coll. reported the 10-year retrospective experience of their center: on 44 patients transplanted with active BM or extramedullary disease after MAC, they reported a 2-year OS of 44.4% (95% C.I. 30.2–65.4%) and a 2-year RI of 53.0% (95% C.I. 51–55%). Grade 3–4 GvHD at day +100 was 15.8%, 2-year severe chronic GvHD was 25.3%, and 100-day and 2-year NRM were 13.8% and 26.7%, respectively. The main cause of death was once more leukemia relapse, with a 1-year RI of 39.4% and a median OS after relapse of only 4.5 months despite reinduction attempts by DLI, chemotherapy, and, possibly, a second allogeneic HSCT [19].Finally, Shimomura and coll. reported a large experience on 707 patients transplanted with active AML after RIC. The 5-year OS and PFS were 22.0% (95% C.I. 8.5–25.7) and 18.8% (95% C.I. 15.6–22.2), respectively, with relapse as the main cause of death (5-year RI: 53.6%, 95% C.I. 49.7–57.4). A high NRM was observed (27.5%, 95% C.I. 24.0–31.2). Male sex, poor PS, cytogenetic risk and the amount of blasts in peripheral blood were independent prognostic variables linked to worse survival [21].

In more recent years, several advancements in the conditioning of HSCT, as well as in the prophylaxis of GvHD and in the supportive care and infection control practices, have allowed physicians to push forward the age limit of allogeneic HSCT, which now includes *selected* elderly patients up to age 75 years and above. A recent EBMT registry study focused on the outcome of HSCT performed in fit elderly people (i.e., ≥70 years) with persistent AML [20]: 360 patients (median age 72 years, range 70–79) transplanted in the years 2010–2021 were analyzed. Donors were MRD (*n* = 58), MUD (*n* = 228), and haploidentical (*n* = 74). All patients had active disease at transplantation, the majority (59.2%) as the result of PIF, and the rest (40.8%) as refractory relapse. Considering groups according to donor type, survival was best with MRD, due to substantially equivalent RI as with MUD or haploidentical donors, but much lower NRM. In fact, 2-year OS and LFS were 62.4% (95% C.I. 47.2–74.3) and 47.6% (95% C.I. 33.1–60.8) for MRD, 43% (95% C.I. 35.8–49.9) and 37.5% (95% C.I. 30.7–44.4) for MUD, and 25.9% (95% C.I. 15.8–37.2) and 26.5% (95% C.I. 16.3–37.8) for haploidentical donors. On the other hand, 2-year RI was 34.9% vs. 30.2% vs. 29.6%, and NRM was 17.5% vs. 32.2% vs. 43.9% among the three donor groups, respectively. Using a more strict composite survival parameter (i.e., GvHD-free, relapse-free survival, GRFS), MRD still appeared the most favorable group: in fact, GRFS was 35.3% (95% C.I. 22.3–48.5) for MRD vs. 29.6% (95% C.I. 32.2–36.2) for MUD vs. 19.2% (95% C.I. 10.7–29.6) for haploidentical transplants [20].

In summary, these studies collectively show the following:a.The main cause of treatment failure in patients transplanted with refractory AML is still disease progression, also when analyzing data from more recent years, with RI ranging around 40–50% in most studies [7,8,9,10,11,12,13,14,15,16,17,18,19,20,21];b.NRM is significantly higher than in the case of HSCT performed in patients in remission, up to 40% after MAC in some reports [7,8,9,10,11,12,13,14,15,16,17,18,19,20,21,22,23,24]; this is particularly true in patients aged ≥70 years [20];c.The intensity of conditioning provides better early disease control in some studies but rarely translates into better long-term OS [11,16]; this is similar to what is observed in studies comparing MAC and RIC in patients transplanted in remission, where better disease control by MAC is usually offset by an increased NRM. Among large studies, only the one by Scott and coll. [25] observed an advantage by MAC over RIC (in terms of PFS, not OS), while many others did not [26,27,28]. It is likely that most of the benefit provided by allogeneic HSCT is due to the transfer of alloreactive immune cells against leukemia (i.e., the GvL effect) more than to the intensity of conditioning;d.Patients transplanted with PIF AML showed better survival than patients with refractory relapsed disease due to a lower risk of relapse [14,16,18,29]. Additional attempts to induce remission yielded poorer results than proceeding directly to allogeneic HSCT;e.Although various prognostic scores based on pre-transplant and post-transplant variables have been developed, none has been universally adopted in clinical practice. Nevertheless, the presence of adverse genetics and the amount of residual BM disease, together with the persistence of extramedullary disease, seem to be the most predictive factors [10,11,13]. It has been consistently proven that the minority of patients that do not present with adverse prognostic factors may still experience significant rates of long-term survival, in the range of 30–40% [8,9,18,21].

Furthermore, although the rationale of using allogeneic HSCT is mainly to provide a kind of immunotherapy against chemoresistant AML, details about the withdrawal of immunosuppression following HSCT and about the administration of DLI as prophylaxis of the relapse (pDLI) are rarely available in retrospective studies. This situation is marginally better when analyzing the available prospective trials. In fact, also in this case the heterogeneity of inclusion criteria, transplant regimens, GvHD prophylaxis, in vivo T-cell depletion, and pDLI administration hamper the possibility to critically compare the different results. This ultimately leads to reported OS ranging from 19% to 70%, with LFS ranging from 19% to 62%, RI from 20% to 54%, and NRM from 20% to 54%. In Italy, the prospective GANDALF01 trial by the cooperative *“Gruppo Italiano per il Trapianto di Midollo Osseo”* (GITMO) [17] failed to improve long-term survival with HSCT performed after MAC based on two alkylating agents, with a 5-year PFS and OS ranging around 20% overall [17].

Despite these limitations, some conceptual notes can still be taken also from the available prospective trials as follows:a.A vast majority of patients achieved at least a transitory remission of their leukemia and the successful engraftment of the transplant, proving the feasibility and effectiveness of HSCT also in the presence of active disease;b.No universal prognostic score for these patients can be derived also from these studies;c.It appears useless to administer more than two induction attempts in refractory patients, as a higher number of chemotherapy courses prior to HSCT has consistently been shown to adversely impact survival [22]. This is in line with those studies that show a better outcome for PIF AML patients as compared to relapsed refractory ones [14,16,18].

## 4. Allogeneic HSCT Performed After Intensified Sequential Conditioning

### 4.1. Sequential Conditioning by Fludarabine, High-Dose Cytarabine, and Amsacrine (FLAMSA)

It has been repeatedly proven, in vitro as well as in vivo, how chemoresistance of AML blasts can be overcome by intensifying the dose or pharmacokinetics of the drugs involved. It is therefore conceivable that, in patients, an augmented sequential conditioning might improve disease control, reduce the risk of early relapse after HSCT and, thus, provide the time window needed for a more effective GvL effect to develop. Unfortunately, a higher risk of infectious complications and NRM may counterbalance these benefits. Table 2 lists the main published studies testing intensified sequential conditioning in patients transplanted with active AML.

This concept was originally developed using a preparative regimen based on fludarabine (30 mg/m^2^ qd for 4 days), high-dose cytarabine (2 gr/m^2^ qd for 4 days), and amsacrine (100 mg/m^2^ qd; overall: the FLAMSA regimen) as pretreatment of a later RIC based originally on TBI (4 Gy, day -4) and cyclophosphamide (40 mg/kg qd for MRD, 60 mg/kg qd for matched unrelated and mismatched donors, days -3 and -2), with HSCT performed during the aplastic phase [30,31]. GVHD prophylaxis was based on anti-thymocyte serum (ATG, 10 mg/kg qd for MRD, 20 mg/kg qd for unrelated and mismatched donors, days -3, -2, and -1), cyclosporin-A (CSA) (therapeutic range 130–200 ng/mL, from day -1), and mycophenolate mofetil (MMF) (from day 0 to day +50). Enrollment criteria included patients affected by PIF AML after two attempts of remission induction by intensive chemotherapy, patients relapsing within 6 months from initial response, patients with AML relapsed and refractory to high-dose cytarabine, and second or later relapses. For 103 patients included in the first report [31], 4-year OS was 32%, LFS 30%, 1-year RI 28.7%, and 1-year NRM 17.2%, all remarkable achievements in this particularly unfavorable series. Notably, patients receiving ≤2 chemotherapy cycles prior to HSCT experienced strikingly better survival (i.e., 60% at 4 years) [31]. The administration of pDLI was an integral part of the protocol and was planned from day +120 from HSCT in the absence of active GVHD, infections, and ongoing immunosuppression (the latter planned for suspension at day +90). Despite this, DLI was actually performed only in one quarter of all patients (23%, in 17/73 patients treated in only two of the participating centers), often with a delay: in fact, the median time from HSCT to first DLI was reported as 159 days (range: 120–284 days). This led to only 9 out of 17 patients receiving three full doses (i.e., 1 × 10^6^, 1 × 10^7^ and 2 × 10^7^ CD3+ cells/kg) [31]. Despite this, better results were observed in patients that actually received DLI as compared to the rest of the group, with 14 out of 17 alive in continuous CR with a median follow-up of 31.5 months (range 15.5–56), and 3-year OS 87%. Grade-III acute GvHD developed in two and chronic GVHD in five of these patients [31]. In 2020, the authors that developed this program published a detailed review updating the original results and focusing on potential strategies to implement it for unfit and elderly patients [55].

Variants of this protocol have been developed using alternative RIC regimens, such as the following: (1) fludarabine (30 mg/m^2^ qd, days -5 and -4) + busulfan (0.8 mg/kg bid at day -6, 0.8 mg/kg *quater in die* at days -5 and -4); (2) cyclophosphamide 120 mg/kg + TBI 4 Gy; and (3) busulfan 6.4 mg/kg + cyclophosphamide 120 mg/kg. Schneidawind and coll. [32] *retrospectively* analyzed the results of these three strategies on 62 patients with relapsed or refractory AML: 2-year OS and EFS were 39% and 26%, respectively, with 2-year RI of 52% and NRM of 22%. No statistical difference in terms of OS was observed (2-year OS of BuFlu 46% vs. TBI-Cy 32% vs. BuCy 44%, *p* = NS). Aiming at reducing the overall toxicity of this approach, a slightly different protocol implementing Treosulfan (10 g/m^2^ qd from day -6 to -4) as a substitute for TBI (4 Gy) was used in the series reported by Holtick and coll. [33] This series also included 130 patients undergoing FLAMSA-RIC while in remission (CR1, *n* = 47; CR2, *n* = 26), besides 57 patients with active AML. OS and DFS of the entire cohort were 45% and 40%, respectively, with 4-year RI at 40% and 4-year NRM at 20%, statistically equivalent in the three groups [33]. Other studies with the FLAMSA regimen confirmed 3-year OS ranging from 15% to 30% [34,35] using various different regimens as RIC.

Although all these studies seem to point out a superiority of the FLAMSA approach over conventional conditioning, two recent reports by the EBMT have partially challenged this notion: in the first one [43], FLAMSA-RIC was compared to treosulfan+fludarabine (TF) or thiotepa–busulfan–fludarabine (TBF), with statistically comparable results among the three groups (FLAMSA + RIC vs. TF vs. TBF) in terms of 2-year RI (53% vs. 46% vs. 54%, *p* = 0.33), NRM (20% vs. 26% vs. 24%, *p* = 0.24), LFS (27% vs. 29% vs. 22%, *p* = 0.28), GRFS (20% vs. 23% vs. 13%, *p* = 0.15), and OS (34% vs. 37% vs. 24%, *p* = 0.10). In the second one [44], the FLAMSA-RIC approach (where RIC was administered either as chemotherapy or by TBI) was compared to a conventional MAC (either TBI-Cy, *n* = 318, or BuCy-based, *n* = 258) in younger patients: the FLAMSA-RIC approach performed by chemotherapy showed better OS (50% vs. 36% vs. 34%, *p* = 0.03) than both the FLAMSA-TBI and the MAC cohorts, but not as a result of better disease control (RI being statistically equivalent in all groups, range 51–56%, *p* = 0.86), but rather as the consequence of reduced 2-year NRM (7% vs. 18% in the FLAMSA-TBI and 16–19% in the MAC cohorts, *p* = 0.04) [44]. No patient received DLI as prophylaxis, but a minority of patients with persistent post-transplant measurable residual disease received pre-emptive DLI (8.3% in the FLAMSA-CT, 15.5% in the FLAMSA-TBI, and 6.6% in the MAC groups).

### 4.2. Sequential Conditioning with Other Chemotherapeutic Regimens

Sequential intensified conditioning can also be performed with other chemotherapeutic regimens consisting of non-crossresistant antileukemic drugs. An example is the study by Dulery and coll. reporting on 72 patients undergoing allogeneic HSCT because of various hematological disorders (only 44 of which being refractory AML). The conditioning regimen consisted of Thiotepa (10 mg/kg qd, days -15 to -10), Etoposide (400 mg/m^2^ qd, days -15 to -10), and cyclophosphamide (1600 mg/m^2^ days -15 to -10), followed by a BuFlu RIC. Results are relevant, with the *caveat* of only 61% of the series being actually patients with refractory AML: in fact, 2-year OS varied from 54.7% (in the case of haploidentical HSCT) to 49.2% (MRD) and 37.9% (MUD), and better outcomes were observed in the case of haploidentical HSCT also with regard to GVHD incidence (grade II-IV 11% vs. 41.4% with MUD, *p* < 0.001) and 2-year GFRS (44.4% vs. 10.3%, *p* = 0.022).

More recent approaches have been using CLAG, a regimen combining cladribine and high-dose cytarabine, as pre-conditioning therapy [42]: Wang and coll. reported a 1-year OS of 69.4% and LFS of 52.9% in 36 patients undergoing CLAG followed by BuCy conditioning. A slightly different variant, adding mitoxantrone (10 mg qd, days -15, -14, and -13) to CLAG, was tested by Sun and coll. [46], with the achievement of remission in 23 out of 24 patients and a 2-year NRM of only 9.1%. Despite this, 2-year OS and LFS were significantly lower (i.e., 61.4% and 59.4%, respectively) due to later relapses in 8 out of 23 cases (at the median time of 4.7 months). Moreover, the CLAG regimen was also combined with TBI (4 Gy qd, days -11 to -9) [50]: in a study involving 70 patients with active AML, 3-year OS and RFS were 46% and 38.5%, respectively, with RI 38.6% and 11.6%. Finally, Xiao and coll. reported on 23 patients transplanted with active AML after cladribine alone (5 mg/m^2^ for 5 consecutive days, days from -9 to -5) followed by BuCy MAC: 2-year OS was 64% and EFS 53.1% [51].

Clofarabine has also been combined with intermediate-dose cytarabine as sequential pre-treatment before conditioning and HSCT: Mohty and coll. reported on 24 patients treated with CL-A (clofarabine 30 mg/m2 + cytarabine 1 g/m^2^ for 5 consecutive days) before BuCy and allogeneic HSCT. All patients were scheduled to receive pDLI starting from day +120, in the absence of active GVHD and immunosuppression (stopped at day +90). 2-year OS and LFS of 38% and 29%, respectively, were reported, with 2-year RI of 54.2% and NRM of 12%. Although pDLI did have a positive impact, only a minority of patients (6 out of the 19 alive at +120 days) could undergo the planned treatment [37].

Another recent iteration of the concept of sequential conditioning involves the use of high-dose melphalan, a highly myelotoxic alkylating agent that was previously uncommon in the repertoire for AML [41,47,52,53]. In their study, Steckel and coll. retrospectively reported on 292 patients affected by relapsed (*n* = 51) or refractory (n = 97) AML and transplanted by means of TBI-based (8 Gy) or treosulfan-based dose-adapted conditioning after high-dose melphalan (140 mg/m^2^). Median age was 56 years, with patients transplanted up to the age of 74 years (range 17–74), all with active disease as a consequence of primary induction failure (*n* = 144), failure of rescue therapy after relapse (*n* = 97), or direct transplantation at relapse without a reinduction attempt (*n* = 51). Three-year OS was 34%, 29%, and 41% in the three groups, respectively, with a nonsignificant trend towards better OS for patients with relapsed AML and a peak 3-year OS of 51% in patients transplanted with leukemic BM infiltration <20%. One year, RI was 34% (95% C.I. 28–41) and NRM was 36% (95% C.I. 31–42). Higher age, transplantation with mismatched donors, and high disease burden independently predicted worse survival. Once more, patients with PIF showed a better outcome when transplanted earlier (i.e., after ≤2 courses) [41]. In another study, Sockel and coll. reported a 4-year OS and EFS of 43% (95% C.I. 36–52%) and 34% (95% C.I. 28–43%), respectively, for 173 patients transplanted with active disease after high-dose Melphalan and sequential RIC-HSCT. Blast count > 20% was associated with worse OS (HR = 1.80, *p* = 0.026) and EFS (HR = 1.93, *p* = 0.009), but not when HSCT was performed during first-line therapy [47]. A third study by Ronnacker and coll. observed a 3-year RFS of 47% (95% C.I.: 40–55%) and OS of 52% (95% C.I. 45–60%) in 176 patients transplanted after high-dose melphalan administered 11 days before HSCT. Again, they reported a significantly lower NRM (HR 0.46, *p* = 0.017) for those patients achieving blast clearance at day 5 after Melphalan, even though that did not translate into an eventual advantage in RFS (HR 071, *p* = 0.09) and OS (HR 0.75, *p* = 0.19). Mismatched donors, older age, adverse genetic risk factors, and higher comorbidity scores were associated with poorer prognoses [52]. Finally, the same group reported a 3-year RFS of 40% (95% C.I. 31–51%) and OS of 44% (95% C.I. 35–55%), with 3-year RI and NRM of 28% (95% C.I. 19–37%) and 32% (95% C.I. 23–41%), respectively, in 103 elderly (i.e., ≥55 years old) patients transplanted with active AML. Transplantation from a mismatched donor was a major risk for worse OS (HR 3.03, *p* < 0.001) and NRM (HR 2.86, *p* = 0.005), while a higher leukemic burden (i.e., ≥20%) was associated with increased RI (HR 3.0, *p* = 0.032) [53].

Pre-conditioning with high-dose melphalan and RIC was also used with haploidentical donors [54,55,56,57]: after retrospectively comparing 21 matched pairs of elderly patients (i.e., ≥50 years old) transplanted after either fludarabine–cyclophosphamide–melphalan (the latter at 110 mg/m^2^) or fludarabine–cyclophosphamide–treosulfan (the latter at 30 gr/m^2^), Fraccaroli and coll. found a strikingly lower RI in the melphalan group (0% vs. 24%, *p* = 0.006), which was partially counterbalanced by a higher NRM (33% vs. 10%, *p* = 0.05), leading to comparable OS and LFS at 2 years (OS 66% and LFS 66%, *p* = 0.80 and *p* = 0.57, respectively). Time-to-engraftment and incidence of acute GvHD did not statistically differ in the two groups [54].

Other types of sequential conditioning include the following:*High-dose Decitabine:* Lv and coll. recently published the results of a multicenter prospective phase II study on 70 patients affected by active AML (in 74.3% of cases after failure of venetoclax) at the time of HSCT using high-dose decitabine (i.e., 400 mg/m^2^) as pre-treatment before BuCy MAC and allogeneic HSCT. Prophylactic DLIs were an integral part of the trial protocol. The regimen appeared overall well tolerated, and results were relevant: in fact, 2-year OS of 58.6% (95% C.I. 47–73%) and LFS of 55% (95% C.I. 43.5–69.4%) were reported, with 2-year RI of 29.6% (95% C.I. 18.4–41.7%) and 2-year NRM of 15.5% (95% C.I. 7.8–25.5%) [58].*Bcl-2 antagonists (Venetoclax):* the addition of venetoclax (200–400 mg qd starting from day -8 for 6–7 days) to a Busulfan/Fludarabine-based RIC (FluBu2) in a study on 22 patients affected by high-risk AML or myelodysplastic syndromes (MDS) (with only 5 AML transplanted not in remission) did not impair the engraftment rate nor induce excessive GvHD. Median PFS was 12.2 months, and median OS was unreached at follow-up 14.7 months (range: 8.6–24.8) [59]. A later expansion of this study also demonstrated the feasibility of the maintenance with venetoclax (400 mg qd, days 1–14/cycle) and azacitidine (36 mg/m^2^ i.v. days 1–5/cycle) after HSCT, with the most common grade 3–4 adverse events being leukopenia, neutropenia, and thrombocytopenia, and all infections being grade 1–2. No significant difference in T-cell reconstitution, but a delay in B-cell expansion was observed as compared to patients not [undergoing maintenance. Reported 2-year OS and PFS were 67% and 59%, with RI 41% and no death in remission (i.e., NRM 0%) [60].*Radio-immunoconjugates:* a phase-3 prospective randomized trial comparing the treatment with an anti-CD45 radio-immunoconjugate (i.e., 131I-apamistamab) vs. conventional care, in both cases followed by allogeneic HSCT, in elderly (>55 years) AML patients (the SIERRA trial) has recently shown favorable results for the experimental arm in all patient subsets and conditions [28]. Nevertheless, the well-known caveats and limitations linked to the use of radio-immunoconjugates will probably limit the diffusion of this procedure to most transplantation centers.

## 5. Donor Selection

### 5.1. HLA-Matched Related and Unrelated Donors, Haploidentical, and Other Alternative Donors

The historical difference in results between allogeneic HSCT using matched related or unrelated donors has markedly reduced in recent years, thanks to relevant progress made in HLA-matching (i.e., genomic instead of serologic HLA typing, broader donor pools), conditioning regimens (i.e., the introduction of several RICs), supportive care, and infection control practices [16,61]. Most recent series on patients transplanted with AML in remission show no difference in results based on donor type when comparing MRD, MUD, or even haploidentical donors [62,63,64].

As allogeneic HSCT controls refractory AML by means of a GvL effect, and this in turn depends on the fine immunological differences between host and donors, the case might be made that alternative donors (i.e., MUD, mMUD, and haploidentical) might be more suited to exploit the immunological differences between donor immune cells and host leukemic blasts. The most relevant biological basis for this hypothesis comes from studies on NK cell alloreactivity made in the context of T-depleted haploidentical HSCT.

Natural killer (NK) cells activate by sensing the lack of HLA class I on target cells by specific receptors, i.e., the killer cell immunoglobulin-like receptors (KIRs) and the NKG2A receptor [65,66,67,68]. After HSCT, newly formed NK cell subsets differ in sensitivity towards HLA-A, -B, -C, and -E, and only a minority actually acquire killing capabilities towards foreign cells as the result of the contemporary lack of inhibitory signals and presence of activating factors [65,67,68]. While only the minority of HLA-A and -B allotypes possessing the Bw4 domain can be recognized by KIRs, all HLA-C molecules react with their respective inhibitory KIRs (i.e., KIR2DL1, 2, and 3) [69]. The HLA-C allotypes, therefore, can be classified into two main groups based on their sequence: in order to elicit a significant NK alloreactivity against leukemic cells, a transplant providing heterozygous HLA-C1/C2 cells against homozygous C1/C1 or C2/C2 is required [70].

An advantage in survival given by KIR mismatch has been demonstrated clinically in the setting of T-depleted haploidentical HSCT [71]. In this regard, a large non-interventional, prospective study by the EBMT analyzed patients undergoing haploidentical transplants between 2012 and 2015 and divided them into those transplanted from NK-alloreactive (*n* = 50) and non-NK-alloreactive donors (*n* = 88). Transplants consisted of ex vivo T-cell-depleted HSCT in 86 cases and unmanipulated HSCT in 52 cases. NK cell alloreactivity did not impact the GRFS of patients with unmanipulated transplants (HR: 1.66, range 0.9–3.1, *p* = 0.10), whereas it proved beneficial in terms of GRFS in T-cell-depleted transplants (HR: 0.6, range 0.3–1.2, *p* = 0.14, interaction *p* < 0.001) as a consequence of a reduced acute and chronic GvHD incidence and reduced NRM [72]. Other studies have also confirmed a beneficial effect of NK cell alloreactivity in the context of T-cell-depleted haploidentical HSCT for AML patients remaining in remission and without active GvHD [73,74].

Unfortunately, these observations could not be replicated in the setting of T-replete HSCT, where the GvL effect mediated by NK alloreactivity is thought to be largely overshadowed by the presence of T-lymphocytes. As a matter of fact, results when studying the effects of a KIR mismatch in HLA-matched T-repleted HSCT have been highly discordant: in HSCT using MRD, the absence of host KIR ligands for the donor’s inhibitory KIRs was found to be either beneficial (in terms of reduced RI and better OS) [75,76] or to have no effect [77]. Conversely, in HSCT using MUD, having a missing KIR ligand was as follows: (1) predictive for increased relapse risk in one study involving AML patients transplanted in HLA-mismatched pairs [78]; (2) associated with reduced RI in patients transplanted with KIR2DS1+/C1+ donors [79] or KIR3DL1+/S1+ grafts [80]; and (3) associated with reduced NRM and acute GvHD in patients transplanted from KIR3DS1+ donors, with no impact on RI [81]. Furthermore, other reports identified a specific donor KIR B gene content score [82] or donor KIR allele (CenB02) [83] as predictive of a decreased risk of relapse, even if these data need further confirmation to draw definitive conclusions.

The role of donor selection based on KIR-mismatch has also been studied in the context of haploidentical T-replete HSCT, with GvHD prophylaxis provided by PTCy administered together with a calcineurin-inhibitor (cyclosporin or tacrolimus) or MTOR-inhibitor (sirolimus) +/- Mycophenolate Mofetil: (1) Symons and coll. reported a beneficial effect on OS by means of reduced RI and NRM in 86 patients transplanted with various lymphoid and myeloid diseases [84]; (2) Bastos-Oreiro and coll. also reported a beneficial effect on event-free survival (EFS) and a reduction in RI by the presence of inhibitory KIR-mismatch in 33 patients transplanted with various hematological conditions [85]; and (3) Wanquet and coll. reported decreased RI in patients transplanted with KIR-L mismatches, even though statistical significance was reached only in patients transplanted with active disease [86]. The positive role of a KIR-mismatch was observed in two other unrelated series [87,88], but disproven in another one [89].

In summary, the benefit of selecting donors based on KIR mismatch is still controversial, with conflicting results in haploidentical HSCT [84,85,87]. This may be explained by the high heterogeneity of the available studies in terms of disease and donor types, immunosuppressive treatments, conditioning regimens, stem cell sources, graft T-cell content, and GvHD prophylaxis.

Besides the selection based on KIR mismatch, there is consistent evidence in the literature on the fact that donor type does not currently impact RI, NRM, and ultimately OS in patients transplanted with refractory disease [61,90,91].

Nevertheless, some considerations may still be relevant as follows:mMUD and haploidentical donors are equally effective as MRD and MUD in terms of RI, LFS, and OS [64], and might also be more effective in some reports, although data are partially inconsistent:a.A study by the EBMT on 1578 patients transplanted with active AML between 2007 and 2014 and divided according to donor type (10/10 MUD *n* = 1111; 9/10 MMUD *n* = 383; haploidentical *n* = 199) showed a statistically equivalent 2-year LFS in the three groups (28% vs. 22.2% vs. 22.8%, respectively, *p* = NS), with nonsignificant differences also in terms of RI, NRM, GFRS, and OS. Interestingly, transplantation with relapsed refractory (as compared to PIF) AML and poorer cytogenetics were independently predictive of poorer survival, but not so donor type [92].b.Konuma and coll. compared the outcome in 5704 elderly patients (i.e., >50 years of age) transplanted with AML either in remission or not, by MRD, 8/8 MUD, 7/8 mMUD, UCB, and haploidentical donors. Overall, donor type did not predict differences in OS, irrespective of disease status at HSCT. Nevertheless, in patients undergoing HSCT while not in remission (*n* = 2663), LFS was significantly better for MUD (HR 0.77, 95% C.I. 0.64–0.93, *p* = 0.005) and UCB (HR 0.76, 95% C.I. 0.65–0.88, *p* < 0.001) as compared to MRD. This was due to lower RI in MUD and UCB. No difference was noted between mMUD (or haploidentical donors) and MRD, as among all donor types in the case of patients transplanted while in remission [91].c.Doppelhammer and coll. reported on 68 patients, 78% of whom were transplanted with active disease and the others with cytogenetically high-risk AML in remission: 3-year LFS was 49% and OS 56%, without significant differences in terms of RI, NRM, and GRFS between MUD and haploidentical donors [93].d.Opposite to these results, a prospective study on 661 patients (275 haploidentical, 246 MUD, and 140 MRD HSCT), all undergoing GvHD prophylaxis by means of PTCy and Tacrolimus (with or without MMF), the haploidentical setting showed reduced OS (HR 2.2, 95% C.I. 1.6–3.0, *p* < 0.001) due to higher NRM (HR 3.2, 95% C.I. 2.0–4.9, *p* < 0.001) as compared to the other two settings [94]. The haploidentical group also showed more infection-related deaths and a higher rate of viral reactivation, grade ≥3 hemorrhagic cystitis, and cardiovascular toxicities, as well as a slower rate of immunological reconstitution. This report, however, is hampered by some limitations: (1) HSCT had been conducted in a variety of hematological diseases; (2) mMUD had not been included; and (3) GvHD prophylaxis had also been partially heterogeneous in relation to the use of MMF, which was practically used only for haploidentical transplants [94].e.Finally, patients transplanted with haploidentical HSCT also showed inferior survival as compared to MRD and MUD in an EBMT study on chemorefractory core-binding factor AML (i.e., characterized by recurrent t(8;21) and inv(16) translocations). Here, the authors reported a 2-year OS and LFS of 53.6% and 42.7%, respectively, but inferior OS (HR 1.79, *p* = 0.003 for MRD and HR 1.64, *p* = 0.004 for MUD) for haploidentical donors as compared to MRD and MUD. Interestingly, patients with t(8;21) experienced higher RI (HR 2.04, *p* = 0.002) as compared to inv(16) patients [95].Haploidentical donors might have a disadvantage at increased age due to higher NRM and poorer GRFS, leading to worse OS [20].Cytogenetic and molecular risk appear more important in predicting outcome than donor choice in most of the available studies [61,90,91,95].HLA heterozygosity after HSCT, deriving from differences in allelic presentation, cellular expression, and up to complete HLA haplotype loss, is relevant to leukemia relapse and a common determinant of immune escape [71,96,97,98]. After haploidentical HSCT, the loss of the mismatched HLA haplotype has been reported in up to one-third of relapsing patients [99]. All these phenomena may be rarer after UCB transplantation [100].

Overall, these studies show how donor selection has been recently widened by the possibility to include haploidentical donors up to an advanced age [61,91]. This is partially tempered by the disadvantage in terms of immune reconstitution, GFRS, and NRM shown by haploidentical transplants [20]. We believe that the search for an alternative donor should be prompted soon after the evaluation of the family and as close as possible to the diagnosis of AML.

The choice of post-transplant immunosuppressive therapy (i.e., ATG vs. PTCy) is highly dependent on the donor type and clinical protocol, PTCy being mainly adopted in the context of T-cell replete haploidentical HSCT and, more recently, 7/8 HLA-matched UD. In the absence of a specific randomized study, it is impossible to draw definitive conclusions on the potential influence of the type of immunosuppressive therapy on the outcome of HSCT in chemorefractory patients independently from other covariates.

### 5.2. Umbilical Cord Blood Transplantation

Umbilical cord blood (UCB) has been poorly investigated as an alternative source of HSC in the setting of adult patients as compared to its much larger use in pediatric patients, due to the well-known intrinsic limitations in the engraftment of UCB and in the immune reconstitution of adult hosts as compared to transplantation with other HSC sources. Nevertheless, some large series have been published on this topic: in the first one [101], 1738 patients receiving transplantation with UCB have been retrospectively compared to 713 patients undergoing HSCT with MRD in the same years (2009–2018), with patients’ characteristics equally balanced in the two groups. The authors observed comparable results in terms of OS and a slightly better PFS for the UCB group; this led to an HR of 0.83 in favor of UCB transplants (95% C.I. 0.69–1.00, *p* = 0.045), despite a higher risk of NRM (HR 1.42, 95% C.I. 1.15–1.76, *p* = 0.001), which was counterbalanced by a better RI (HR 0.78, 95% C.I. 0.69–0.89, *p* < 0.001) [101]. Another registry study by the national Japanese group compared adult patients transplanted with either UCB (*n* = 918) or haploidentical HSCT (*n* = 459) in a 2:1 propensity score matching and also reported statistically comparable results in the two matched cohorts in terms of RI (HR 1.09, 95% C.I. 0.93–1.28, *p* = NS), NRM (HR 0.94, 95% C.I. 0.76–1.18, *p* = NS), and OS (HR 1.02, 95% C.I. 0.89–1.16, *p* = NS) [102].

Conversely, 2963 patients transplanted with active AML with either UCB (*n* = 285), 10/10 HLA-matched UD (*n* = 2001), or 9/10 HLA-matched UD (*n* = 677) in the years 2005–2015 were compared in a third large retrospective registry study by the EBMT [103]. Here, neutrophil engraftment rates at day 60 were 73%, 94%, and 93% in the three groups, respectively (*p* < 0.001), and the rate of CR at day 100 after HSCT was 48%, 69%, and 70%, respectively (*p* < 0.001). In multivariate Cox analysis, 10/10 MUD fared better than UCB transplants in terms of all survival parameters, namely RI (HR 0.7, 95% C.I. 0.6–0.9, *p* < 0.001), NRM (HR 0.6, 95% C.I. 0.4–0.7, *p* < 0.001), GRFS (HR 0.8, 95% C.I. 0.7–0.9, *p* < 0.001), and OS (HR 0.6, 95% C.I. 0.5–0.7, *p* < 0.001). The same was true also for 9/10 HLA-matched UDs, with similar results [103]. As a consequence, unrelated donors appeared overall superior to UCB in the treatment of adult patients with refractory AML, and the use of UCB for transplantation has become increasingly less common in adults in recent years, at least in Europe.

## 6. Post-Transplant Maintenance and Pre-Emptive Treatments

One of the reasons to proceed to allogeneic HSCT in patients with refractory AML is the possibility to establish an immunological platform to exploit in the immunotherapy of leukemia either by DLI or novel immunotherapeutic drugs. Following the demonstration of the GvL effect as the most effective property of allogeneic HSCT against leukemic cells [104,105,106], early reduction in pharmacological immunosuppression has been attempted in the case of patients transplanted with active disease in order to elicit an earlier allogenic immune response against leukemia and improve survival. Overall, these attempts have proven fruitless, as the expected advantage in RI by the improved clearance of residual leukemic blasts has been repeatedly outweighed in terms of OS by the increased mortality due to the facilitation of severe acute GvHD [107]. No guidelines are available on this topic, and published evidence is poor outside of specific clinical trials that often combine the early withdrawal of immunosuppression with the administration of pDLI (such as in the FLAMSA concept) [55]. Therefore, given the scarcity of evidence, together with most transplantologists, we caution against the empirical early withdrawal of immunosuppressive therapy outside a specific clinical trial [108]. This approach can be useful only when dealing with an early immunophenotypic or molecular AML relapse, i.e., detected as measurable residual disease conversion in the post-transplant follow-up, especially after the first 100 days. In fact, in this case the risk of death following hematological relapse supersedes that of facilitating GvHD, while, on the opposite, a quicker tapering of immunosuppression may still elicit a successful GvL effect at the reasonable clinical cost of mild to moderate chronic GvHD, which is not associated with an increased risk of mortality [107].

Furthermore, DLI has been recently established as a standard of treatment in ex vivo lymphodepleted allogeneic HSCT as well as in some cases of T-repleted HSCT. In the latter, DLIs are currently administered as prophylaxis against relapse in the case of high-risk AML [36,106,108] or as pre-emptive therapy in patients with detectable post-transplant measurable residual disease [106,109]. Recommendations on the use of DLI have been issued also in the setting of haploidentical HSCT [110]. Doses vary according to the donor type (ranging from 1 × 10^6^ cells/kg in MRD and MUD to 0.5 × 10^6^ in MMUD or haploidentical/MMRD donors) [110] and ultimately to the risk of facilitating the occurrence of GvHD. Active GvHD, ongoing pharmacological immunosuppression, or active infections are thus fundamental contraindications to the administration of DLI.

The importance of pDLI was highlighted by a recent Indian study confronting two groups transplanted with haploidentical donors after MAC, the first receiving pDLI from day +21 (after a rapid tapering of MMF from day +14 and of CSA from day +60), the other without. Results were strikingly in favor of the group treated with pDLI, with 2-year OS and LFS of 70% and 62% vs. 35% and 25%, respectively [36]. As we already pointed out, pDLI are also an essential part of HSCT made after intensified sequential conditioning such as FLAMSA and others [30,31,37,108]. Nevertheless, in all published studies, only a minority of the patients treated according to these protocols actually managed to undergo repetitive DLI [30,31,37,108], and concerns about the possibility of inducing GvHD remain.

Because of this, the need for pDLI in all patients has been questioned: a recent clinical trial showed, for instance, acceptable results after a sequential program including debulking chemotherapy and RIC by fludarabine and cyclophosphamide (without ATG) also without pDLI. The authors reported 2-year OS of 39%, with 2-year RI of 30%, 1-year NRM of 33%, and no patient relapsing after 2 years [39].

Post-transplant maintenance by hypomethylating agents may be synergistic with pDLI and further ameliorate results: in vitro studies have pointed out how azacitidine and decitabine may improve the GvL effect by increasing the immunogenicity of leukemic blasts by means of epigenetic reactivation of genes expressing embryonic and leukemia-associated antigens [111,112].

## 7. A Special Case: TP53-Mutated AML

TP53 is a tumor suppressor gene encoding for more than 10 isoforms involved in cell signaling, apoptotic mechanisms (by interaction with p21/WAF and Bcl-XL) [113,114] and multiple metabolic pathways (such as the synthesis of fatty acids or the shift to gluconeogenesis and glycolysis) [115,116]. It also has a role as an epigenetic regulator by downregulating DNMT3A and DNMT3B [113] and participating in the modulation of lineage commitment in hematopoiesis [113]. All known TP53 mutations, though highly heterogeneous, affect the DNA-binding domain of the corresponding protein, disabling its ability to sense DNA sequences and therefore its regulatory function [113].

It is well known how TP53-mutated/17p-deleted AML is characterized by intrinsic chemoresistance and a dismal prognosis [117]. Table 3 summarizes the major clinical studies on patients with TP53-mutated AML and MDS.

Overall, these patients achieve remission in only a minority of cases (ranging from 29% to 50%), and usually relapse early, with median OS ranging from 2 to 10 months and 2-year OS < 20% overall [113,128,129,130,131,132]. A recent meta-analysis reported a pooled 2-year OS of 29.7% and a relapse rate of 61.4% [133]. It is therefore unsurprising that most patients with TP53-mutated AML frequently undergo allogeneic HSCT with refractory leukemia.

Recent studies have shown how it is possible to further stratify these patients based on several features. These comprehend the following: (1) the loss of heterozygosity that results in lower levels of functional protein; (2) the presence of co-occurring mutations in genes involved in epigenetic regulation (such as DNMT3A, ASXL1, and TET2, RAS/MAPK, and RUNX1) or in RNA splicing such as SRSF2) [113]; and (3) the co-occurrence of high-risk cytogenetic abnormalities, such as a complex or monosomic karyotype, aneuploidy (including monosomy of 5 and monosomy of 7), and del(5q) or del(3p) [113,126,127,134]. In fact, it has been consistently proven that patients with isolated TP53 at low mutation burden (e.g., VAF ≤ 40%) are associated with a better prognosis, especially when consolidated by allogeneic HSCT; on the opposite, the presence of high VAF value and/or concomitant high-risk cytogenetic features dramatically worsen survival.

This is shown by several studies available in the literature: (1) Short and coll. observed a median OS of 32.3 months in TP53-mutated AML patients with VAF ≤ 40% (as compared to 9.5 months for VAF > 40%, *p* = 0.01) [135]; (2) Lontos and coll., in another series of 240 patients affected by either TP53-mutated AML or MDS, reported 2-year PFS plummeting from 60% to 22% (intermediate risk, 54.6% of the series) and 3% according to VAF levels (threshold: 50%) and the presence of high-risk cytogenetics [127]; (3) a registry study by the EBMT on 179 AML patients transplanted in CR1 reported a 2-year OS ranging from 65.2% to 15.0% (*p* = 0.001), with relapses occurring beyond the usual 2-year mark, depending on the concomitant presence of TP53-mutation and chromosome 17p loss and/or complex karyotype (*n* = 126, 70.4% of the whole series) [126]; (4) finally, another study on MDS and myeloproliferative neoplasms in Japan observed a 2-year OS around 20% for patients presenting with a TP53-mutation together with complex karyotype [122].

The inefficacy of HSCT to prevent relapse may be a consequence of poor disease control prior to transplantation due to the limited tendency of TP53-mutated blasts to undergo apoptosis, but it may also stem from peculiar immunological features [113]. In fact, infiltrations by immune cells in the BM microenvironment are common in these patients [136] and may be correlated with increased PD-L1 expression by TP53-mutated blasts [137] and stronger IFN-a and IFN-g production from their T lymphocytes as compared to healthy donors [138]. A recent trial on MDS/AML patients has also highlighted a peculiar frailty of these patients to infections (including pneumonia and invasive fungal infections), despite similar durations of neutropenia, and after normalization for the lower rate of CR [139]. Although confirmation studies are needed, all these features may hamper the GvL effect in patients with TP53-mutated AML as well.

This sparked a sharp debate on the appropriateness of allogeneic HSCT for patients at higher risk, i.e., with high VAF ratios of TP53-mutated genes or with concurrent complex or monosomic karyotype [130]. No evidence-based treatment guidelines for these patients are available, as yet: myeloablative or intensified conditionings do not seem to offer advantages over RIC [118,121,123,126,127,140], and several recent trials with novel drugs have not led, so far, to real progress in the field.

All that considered, we still believe the presence of a TP53 mutation not to be an absolute contraindication to proceed to allogeneic HSCT, as long-term results have been in the range of 30–35%, especially in the subgroup with lower VAF and without concomitant high-risk genetic features [127,135]. In a study by the MD Anderson, only the presence of CR/CRi on day +100 post-transplantation and the occurrence of chronic GvHD predicted prolonged OS, indirectly supporting the importance of the GvL effect (usually tied to the incidence of chronic GVHD) in providing the best chance of long-term disease control [141]. Chronic GvHD was associated with slightly better OS also in the other two studies [142,143], in which the authors observed median OS ranging from 33.7 months to 7.0 months in transplanted vs. untransplanted patients [143]. We nevertheless support the relevance of the prognostic stratification of these patients by means of VAF measurement and cytogenetic analysis, as suggested by others [126,127,130], although, in the absence of a specific clinical trial, we argue that allogeneic HSCT still provides the best therapeutic option also in the case of high-risk patients, given the possibility of additional post-transplant immunotherapy [112]. Premature withdrawal of immunosuppression and the administration of pDLI are generally advised, especially if transplants are performed with measurable residual disease.

## 8. Conclusions

Chemotherapy has historically yielded exceptional results in the therapy of AML: a disease once thought incurable can now achieve sustained long-term remission and cure in almost 50% of patients overall [1,2,3]. Despite this, the prognosis of patients failing to respond to induction therapy remains dismal, due to intrinsic or acquired chemoresistance, and often as the result of clonal evolution [144]. Allogeneic HSCT provides the only chance of cure in these patients by means of the immunological counterplay provided by the transplanted healthy alloreactive immune system (i.e., the GvL effect) [104,105,106]. Despite this, results for these patients are unsatisfying also when undergoing HSCT.

As promising approaches to improve these results, we believe sequential conditioning to be useful in patients with lower risk of early NRM, e.g., patients with lower Hematopoietic Cell Transplantation-specific Comorbidity Index (HCT-CI) [145], up to advanced age (i.e., 70 years) in selected fit patients. Proper patient selection, reduced intensity conditioning, and dose reduction also for the pre-conditioning therapy should always be adopted above age 60 years. Performance status and the risk of severe infections, as well as prior infectious history, should also be considered, as patients still recovering from previous infections are at higher risk to suffer from intensified sequential conditioning. Despite this, we believe this risk to be balanced by the possibility to overcome disease resistance and, as such, to facilitate better early-stage engraftment and the achievement of a status of complete remission, all fundamental preconditions for the development of the GvL effect.

Following several reports [14,16,18,22,146], we also do not believe additional chemotherapy to be useful for patients failing two induction attempts, especially when high-risk genetic features are present that suggest poor chemosensitivity. We would rather proceed to allogeneic HSCT as soon as possible, should a matched donor be available either in the family or from the international registry, in order to avoid potential toxicity or the risk of infectious complications that might hamper the chance to perform HSCT. We believe fewer chemotherapy courses explain, at least in part, the advantage in survival shown by several studies comparing PIF patients to refractory relapsed AML [14,16,18]. Novel drugs are warranted, especially immunotherapeutic agents favoring the GvL, given before or after transplantation. Unfortunately, they are unlikely to happen in the foreseeable future.

Finally, even though the treatment of chemorefractory AML remains an unresolved issue, we believe that allogeneic HSCT remains at present the best option to offer our patients outside experimental clinical trials. We hopefully look to the clinical development of the immunotherapy of AML to improve in the near future the dismal prognosis of these highly unfavorable patients.

## Figures and Tables

**Table 1 cancers-17-03285-t001:** Selected clinical studies on patients transplanted with active AML after conventional conditioning.

Authors	N	Study	Median Age	Inclusion Criteria	Donor Type	Conditioning	GvHD Prophylaxis	pDLI	OS	RI	NRM
*Michallet* et al., *2000* [7]	230	*Retrospective*	-	*PIF, untreated and refractory relapse*	MRD/MUD	MAC	CSA + MTX +/− ATG	No	(5 yr) 9–13%	*-*	*(5 yr) 45%*
*Duval* et al., *2010* [8]	2255	*Retrospective*	38	*PIF, untreated and refractory relapse*	MRD/MUD	MAC	FK/CSA +/− MTX (T-cell depletion 13%)	No	(3 yr) 19% (6–42%)	*-*	*(3 yr) 38%*
*Craddock* et al., *2011* [9]	168	*Retrospective*	40	*PIF*	MRD/MUD	MAC/RIC	-	No	(5 yr) 22%	*-*	*-*
*Hemmati* et al., *2014* [10]	131	*Retrospective*	52	*PIF + refractory relapse*	MRD/MUD	MAC/RIC/FLAMSA-RIC	CSA + MTX/MMF	Yes	-	*(5 yr) 48%*	*(3 yr) 26%*
*Liu* et al., *2015* [11]	133	*Retrospective*	40, 30, 21	*PIF + refractory relapse*	MRD/MUD/Haplo	MAC	CSA + MTX +/− MMF Others (GIAC)	No	(3 yr) 40%	*-*	*(3 yr) 19%*
*Nagler* et al., *2015* [12]	852	*Retrospective*	43, 39	*PIF + refractory relapse*	MRD/MUD	MAC	CSA + MTX + ATG	No	(2 yr) 31% and 33%	*(2 yr) 53% and 54%*	*(2 yr) 21% and 17%*
*Todisco* et al., *2017* [13]	227	*Retrospective*	49	*PIF*	MRD/MUD/Haplo/CB	MAC/RIC	T-cell depletion 50%	No	(3 yr) 14%	*(3 yr) 61%*	*(3 yr) 27%*
*Weisdorf* et al., *2017* [14]	2696	*Retrospective*	49–52	*CR2 + PIF + refractory relapse*	MRD/MUD/mMUD	MAC/RIC	CSA + MTX +/− ATG	No	(2 yr) 18–21%	*(2 yr) 41–40%*	*(2 yr) 28–25%*
*Poiani* et al., *2021* [15]	2089	*Retrospective*	55	*PIF + untreated and refractory relapse*	MRD/MUD/mMUD	MAC/RIC	CSA + MTX/MMF +/− ATG	No	(2 yr) 22–41%	*(2 yr) 42–61%*	*(2 yr) 21%*
*Nagler* et al., *2022* [16]	3430	*Retrospective*	55	*PIF + refractory relapse*	MRD/MUD/Haplo	MAC/RIC/FLAMSA-RIC	CSA + MTX/MMF +/− ATG (78%) or PTCy (4%)	No	(2 yr) 36%	*(2 yr) 48%*	*(2 yr) 24%*
*Bonifazi* et al., *2022* [17]	101	*Prospective*	54	*PIF + refractory relapse*	MUD/Haplo/CB	TBF	ATG-based/PTCy-based	No	(2 yr) 19%	*(2 yr) 53%*	*(1 yr) 35%*
*Yanada* et al., *2023* [18]	6927	*Retrospective*	53	*PIF + refractory relapse*	MRD/MUD/CB	MAC/RIC	FK/CSA-based	No	(5 yr) 23%	*(5 yr) 53%*	*(5 yr) 27%*
*Jiang* et al., *2023* [19]	44	*Retrospective*	-	*PIF + refractory relapse*	MRD/MUD	MAC	CSA-MMF +/− ATG	No	(2 yr) 44.4%	*(2 yr) 53%*	*(2 yr) 26.7%*
*Maffini* et al., *2024* [20]	360 (≥70 yrs)	*Retrospective*	72	*PIF + refractory relapse*	MRD/MUD/Haplo	RIC	CSA-MMF + ATG/PTCy	No	(2 yr) 25.9–62.4%	*(2 yr) 29.6–34.9%*	*(2 yr) 17.5–43.9%*
*Shimomura* et al., *2025* [21]	707	*Retrospective*	53	*PIF + refractory relapse*	MRD/MUD	RIC	FK/CSA-based	No	(5 yr) 22%	*(5 yr) 53.6%*	*(5 yr) 27.5%*

PIF: primary induction failure; CR2: second remission after relapse and successful rescue therapy; MRD: matched related donors; MUD: matched unrelated donors; mMUD: mismatched unrelated donors; MAC: myeloablative conditioning regimens; RIC: reduced intensity conditioning regimens; FK: tacrolimus; CSA: cyclosporine A; MTX: methotrexate; MMF: mycophenolate mofetil; ATG: anti-thymocyte globulin; PTCy: post-transplantation cyclophosphamide; GIAC: granulocyte-intensified immunosuppression by antithymocyte globuline combination; pDLI: prophylactic donor lymphocyte infusion; RI: relapse incidence (cumulative function); OS: overall survival; NRM: non-relapse mortality; yr: year.

**Table 2 cancers-17-03285-t002:** Selected clinical studies on patients transplanted with active AML after sequential intensified conditioning.

*Authors*	N	*Study*	Median Age	*Inclusion Criteria*	Donor Type	Conditioning	GvHD Prophylaxis	pDLI	OS	*RI*	*NRM*
*Schmid* et al., *2005* [30]	75	*Prospective*	52	*PIF, early and refractory relapse, ≥2nd relapse, secondary AML/MDS*	MRD/MUD	FLAMSA + TBI(4 Gy)Cy	CSA + MMF + ATG	Yes	(2 yr) 42%	*(2 yr) 20%*	*(1 yr) 33%*
*Schmid* et al., *2006* [31]	103	*Prospective*	51	*PIF, early and refractory relapse, ≥2nd relapse*	MRD/MUD	FLAMSA + TBI(4 Gy)Cy	CSA + MMF + ATG	Yes	(2 yr) 40%	*(2 yr) 37%*	*(1 yr) 17%*
*Schneidawind* et al., *2013* [32]	62	*Retrospective*	-	*PIF + relapse*	MRD/MUD/mMUD/Haplo	FLAMSA + BuFlu/BuCy/TBICy	CSA + MMF + ATG	No	(2 yr) 31–46%	*-*	*(2 yr) 20–26%*
*Holtick* et al., *2015* [33]	130	*Retrospective*	-	*CR1 + CR2 + PIF + relapse*	MRD/MUD/mMUD/Haplo	FLAMSA + BuFlu/ TreoFlu	CSA + MMF + ATG	Yes	(4 yr) 45%	*(4 yr) 40%*	*(4 yr) 20%*
*Pfrepper* et al., *2016* [34]	44	*Retrospective*	52	*PIF + relapsed refractory*	MRD/MUD/mMUD	FLAMSA + TBI(4 Gy)Cy	CSA + MMF	Yes	(3 yr) 15%	*(3 yr) 69%*	*(3 yr) 18%*
*Middeke* et al., *2016* [35]	84	*Prospective*	61	*PIF + relapse*	MRD/MUD/mMUD	CL-A + CL-HD-MEL	CSA + MMF + ATG (mMUD only)	No	(2 yr) 43%	*(2 yr) 26%*	*(2 yr) 23%*
*Jaiswal* et al., *2016* [36]	41	*Prospective*	26	*PIF + refractory relapse*	Haplo	BuFlu-HD-MEL	CSA + MMF + PTCy	Yes	(18 mo) 53%	*(1 yr) 43%*	*(1 yr) 19%*
*Mohty* et al., *2017* [37]	24	*Prospective*	47	*PIF + persisting hypoplasia*	MRD/MUD/mMUD	CL-A + BuCy	CSA + MMF (MUD/mMUD only) + ATG	Yes	(2 yr) 38%	*(2 yr) 54.2%*	*(2 yr) 12%*
*Ringden* et al., *2017* [38]	267	*Retrospective*	51	*PIF + refractory relapse*	MRD/MUD	FLAMSA + TBI/Cy/Bu-based/HD-MEL	CSA + MTX/MMF + ATG	No	(3 yr) 30%	*(3 yr) 48%*	*(3 yr) 26%*
*Davies* et al., *2018* [39]	47	*Prospective*	53	*PIF + early and refractory relapse*	MRD/MUD	Dauno-Ara-C + FluCy	CSA + MTX	No	(2 yr) 39%	*(3 yr) 30%*	*(1 yr) 35%*
*Dulery* et al., *2018* [40]	72	*Retrospective*	54	*PIF + refractory relapse*	MRD/MUD/Haplo	TEC + BuFlu	CSA + MMF + ATG	Yes	(2 yr) 57%	*(2 yr) 38%*	*(2 yr) 24%*
*Steckel* et al., *2018* [41]	292	*Retrospective*	56	*PIF + untreated relapse*	MRD/MUD	HD-MEL+ TBI(8Gy)Flu/TreoFlu	CSA + MTX/MMF + ATG	No	(3 yr) 34%	*(1 yr) 34%*	*(1 yr) 36%*
*Wang* et al., *2018* [42]	36	*Prospective*	26.6	*PIF + refractory relapse*	MRD/MUD/Haplo	CLAG + BuCy	CSA + MTX + MMF +/− ATG	No	(1 yr) 69.4%	*-*	*-*
*Saraceni* et al., *2019* [43]	856	*Retrospective*	51, 58	*PIF + refractory relapse*	MRD/MUD	FLAMSA + TBI-Bu/TreoFlu/TBF	CSA + MTX/MMF + ATG	Yes	(2 yr) 34% and 37% and 24%	*(2 yr) 53% and 46% and 54%*	*(2 yr) 20% and 26% and 24%*
*Rodriguez-Arboli* et al., *2020* [44]	1018	*Retrospective*	39	*PIF + refractory relapse*	MRD/MUD	FLAMSA + TBI-based/chemo-based MAC	CSA + MTX/MMF + ATG	No	(2 yr) 36% and 50%and 33%	*(2 yr) 55% and 53% and 51%*	*(2 yr) 18% and 7% and 19%*
*Le* Bourgeois et al., *2020* [45]	131	*Retrospective*	52	*PIF + refractory relapse*	MRD	CL-A + BuCy	CSA + MTX/MMF + ATG	No	(2 yr) 38%	*(2 yr) 45%*	*(2 yr) 35%*
*Sun 2021* [46]	24	*Prospective*	32	*PIF + refractory relapse*	MRD/MUD/Haplo	CLAG-M + BuCy	CSA + MTX + MMF + ATG	No	(2 yr) 61.4%	*(2 yr) 34.8%*	*(2 yr) 9.1%*
*Sockel* et al., *2022* [47]	173	*Retrospective*	56	*refractory relapse*	MRD/MUD/Haplo	CL-A + Flu-HD-MEL/CL-HD-MEL	CSA + MTX/MMF + PTCy	No	(4 yr) 43%	*(4 yr) 30%*	*(4 yr) 36%*
*Guijarro 2022* [48]	140	*Retrospective*	55	*PIF + refractory relapse*	MRD/MUD/Haplo	FLAG-Ida + HD-MEL	CSA + MTX/MMF + ATG/PTCy	No	(5 yr) 25%	*(5 yr) 30%*	*(5 yr) 45%*
*Weller 2022* [49]	114	*Retrospective*	60	*PIF + refractory relapse*	MRD/MUD/Haplo	FLAMSA + RIC	CSA + MTX/MMF + ATG/PTCy	Yes	(2 yr) 45%	*(2 yr) 41%*	*(2 yr) 27%*
*Fei* et al., *2023* [50]	70	*Retrospective*	-	*PIF + refractory relapse*	MRD/MUD/Haplo	CLAG + TBI/TBI + Cy	CSA + MMF + ATG	No	(3 yr) 46%	*(1 yr) 38.6%*	*(1 yr) 11.6%*
*Xiao* et al., *2024* [51]	23	*Retrospective*	33	*PIF + refractory relapse*	MRD/MUD/mMUD	Cladribine + BuCy	FK + MTX + MMF + anti-CD25 + ATG	No	(2 yr) 64%	*(1 yr) 13%*	*(1 yr) 13%*
*Ronnacker* et al., *2024* [52]	176	*Retrospective*	61	*PIF + refractory relapse*	MRD/MUD/mMUD	HD-MEL + TBI(8Gy)Flu + BuFlu + TreoFlu	FK/CSA + MMF + ATG	No	(3 yr) 52%	*(3 yr) 27%*	*(3 yr) 26%*
*Ronnacker* et al., *2024* [53]	103 (≥55 yrs)	*Retrospective*	67	*PIF + refractory relapse*	MRD/MUD/mMUD	HD-MEL + BuFlu	FK/CSA + MMF + ATG	No	(3 yr) 44%	*(3 yr) 28%*	*(3 yr) 32%*
*Fraccaroli* et al., *2024* [54]	42 (≥50 yrs)	*Retrospective (match-paired)*	65	*PIF + refractory relapse*	MRD/MUD	FluCy + HD-MEL/Treo	FK/CSA-based	No	(2 yr) 66% (both)	*(5 yr)* 0% vs. 24%	*(5 yr)* 33% vs. 10%

PIF: primary induction failure; CR1: first complete remission; CR2: second complete remission; MRD: matched related donors; MUD: matched unrelated donors; mMRD: mismatched related donors; mMUD: mismatched unrelated donors; Haplo: haploidentical donors; MAC: myeloablative conditioning regimens; RIC: reduced intensity conditioning regimens; TEC: thiotepa (5 mg/kg), Etoposide (400 mg/m^2^), cyclophosphamide (2 gr/m^2^); HD-MEL: high-dose melphalan (100–140 mg/m^2^, depending on the study); CL-A: clofarabine + high-dose cytarabine; CL: clofarabine; FK: tacrolimus; CSA: cyclosporine A; MTX: methotrexate; MMF: mycophenolate mofetil; ATG: Anti-thymocyte globulin; PTCy: post-transplantation cyclophosphamide; pDLI: prophylactic donor lymphocyte infusion; RI: relapse incidence (cumulative function); OS: overall survival; NRM: non-relapse mortality; yr: year; mo: months.

**Table 3 cancers-17-03285-t003:** Selected clinical studies on patients transplanted with TP53-mutated AML.

*Authors*	N	*Study*	Median Age	*Inclusion Criteria*	Patients Transplanted with TP53^mut^ or 17p abn	Subset with High-Risk Cytogenetics	Conditioning	OS	*Median OS*	*NRM*
*Middeke* et al., *2014* [118]	201	*Retrospective*	54	*AML*	201	Monosomal: 77 Complex: 180	MAC: 35% RIC: 52% NMA: 9%	(5 yr) 12%	*8.0 mo*	*14–40%*
*Middeke* et al., *2016* [119]	97	*Prospective (3 trials)*	51	*AML*	40	High-risk: 40	MAC/RIC	(3 yr) 10% *(high-risk)*	*-*	*-*
*Luskin* et al., *2016* [120]	112	*Retrospective*	55.5	*AML*	9	High-risk: 6	MAC: 69% RIC: 31%	All relapsed	*1.6–18.6 mo*	*-*
*Lindsley* et al., *2017* [121]	1514	*Retrospective*	84% >40 yrs	*MDS*	289	-	MAC: 52% RIC: 38% NMA: 9%	(3 yr) 20%	*8.4 mo*	*-*
*Yoshizato* et al., *2017* [122]	797	*Retrospective*	-	*MDS + sAML*	98	Complex: 86	MAC: 65.2% RIC: 34.8%	(3 yr) 10% *(high-risk)*	*4.8 mo*	*-*
*Poiré* et al., *2017* [123]	125	*Retrospective*	54	*AML*	125	Monosomal: 86 Monosomy 5/5q-: 58	MAC: 41% RIC: 59%	(2 yr) 16–28%	*10.5 mo*	*(2 yr) 15%*
*Najima* et al., *2018* [124]	120 (all active AML)	*Retrospective*	51	*AML*	23	Monosomal: 11	MAC/RIC	(2 yr) 27.3%	*-*	*(2 yr) 32.5–41.7%*
*Grob* et al., *2022* [125]	2200	*Prospective (4 trials)*	62	*MDS + AML*	230	High-risk: 112	MAC/RIC	(3 yr) 10%	*10.0 mo*	*-*
*Loke* et al., *2022* [126]	179	*Retrospective*	60.2	*AML*	179	High-risk: 126	MAC: 34.9% RIC: 65.1%	(2 yr) 24.6% *(high-risk)*	*9.5 mo*	*(2 yr) 22%*
*Lontos* et al., *2025* [127]	250 (AML: 126)	*Retrospective*	62	*MDS + AML*	250	High-risk: 189	MAC: 59% RIC: 41%	(2 yr) 34%	*9 mo*	*(2 yr) 22%*

AML: acute myeloid leukemia; MDS: myelodysplastic syndromes; MAC: myeloablative conditioning regimens; RIC: reduced intensity conditioning regimens; NMA: non-myeloablative regimens; yr: years; mo: months; OS: overall survival; NRM: non-relapse mortality; yr: year; mo: months.

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
