# Peer review of "Allogeneic Hematopoietic Stem Cell Transplantation as a Platform to Treat Chemorefractory Acute Myeloid Leukemia in Adult Patients"

_cancers, 2025, doi:10.3390/cancers17203285_

Round 1

Reviewer 1 Report

Comments and Suggestions for Authors

The authors aim to conduct a review on the subject (Allogeneic Hematopoietic Stem Cell Transplantation as a Platform to Treat Chemorefractory Acute Myeloid Leukemia).

I consider that the manuscript is generally well written and easy to follow throughout. I think it covers all the key points of interest in the field ( I guess is just about adult patients)( this point should be clarify adding adult patients in the title). However, the manuscript presents some methodological shortcomings. Authors state that they are conducting a review on the topic in caution but they do not specify what type of review is (for instance; systematized, narrative or concise review). I think this is important because there are important differences in their methodology, scope and rigor. For instance, what was the searching strategy? What were the inclusion and exclusion criteria used for papers review?. Authors present several tables with the selected articles but they do not explain how these articles were chosen. 

I believe these two aspects are the weakest points of the manuscript and should be addressed. I think authors should create a method section explaining how the papers were selected and how many of them were reviewed and/or discarded.

Author Response

Reviewer's comments (#1): The authors aim to conduct a review on the subject (Allogeneic Hematopoietic Stem Cell Transplantation as a Platform to Treat Chemorefractory Acute Myeloid Leukemia).

I consider that the manuscript is generally well written and easy to follow throughout. I think it covers all the key points of interest in the field ( I guess is just about adult patients)( this point should be clarify adding adult patients in the title). However, the manuscript presents some methodological shortcomings. Authors state that they are conducting a review on the topic in caution but they do not specify what type of review is (for instance; systematized, narrative or concise review). I think this is important because there are important differences in their methodology, scope and rigor. For instance, what was the searching strategy? What were the inclusion and exclusion criteria used for papers review?. Authors present several tables with the selected articles but they do not explain how these articles were chosen. 

I believe these two aspects are the weakest points of the manuscript and should be addressed. I think authors should create a method section explaining how the papers were selected and how many of them were reviewed and/or discarded.

Reply to Reviewer # 1:

First of all, we wish to thank the Reviewer for his useful comments and appreciation of our work.

We acknowledge our review dealt with adult AML patients from the very beginning, and systematically excluded studies made with pediatric patients. We made it clearer in the revised title (which now reads as “Allogeneic hematopoietic stem cell transplantation as a platform to treat chemorefractory acute myeloid leukemia in adult patients”) as well in the introductory and methodological sections. As requested, we added a methodology section where the search and selection methods used in writing this review have been detailed.

All changes in the Manuscript are highlighted in green.

Reviewer 2 Report

Comments and Suggestions for Authors

This is a very well written and thorough review of the field and touched most of the important issues.  I enjoyed reading it.

There are 3 areas that I believe comments should be emphasized, but a lot of detail does not need to be included.

  1. a comment on the role of cord bloods at present, if any
  2. a comment on the duration of immunosuppression post HSCT that maximizes the chance of no relapse
  3. has the use of ATG/ptCy made any difference with the increased immunosuppression

Author Response

Reviewer's comments (#2): 

This is a very well written and thorough review of the field and touched most of the important issues.  I enjoyed reading it.

There are 3 areas that I believe comments should be emphasized, but a lot of detail does not need to be included.

  1. a comment on the role of cord bloods at present, if any
  2. a comment on the duration of immunosuppression post HSCT that maximizes the chance of no relapse
  3. has the use of ATG/ptCy made any difference with the increased immunosuppression

Reply to Reviewer #2:

First of al, we wish to thank the Reviewer for his/her appreciation of our work and the nice comments.

We tried to answer the questions posed:

  • With regard to umbilical cord blood transplantation: we added to the Manuscript a whole paragraph (i.e. “5.2. Umbilical cord blood transplantation”) and three articles (i.e. Shimomura, Y. et al., Ref. 102; Matsuda, K. et al., Ref. 103; and Baron, F. et al. Ref 104) specifically addressing this issue
  • With regard to early withdrawal of immunosuppressive therapy as a mean to improve outcome by eliciting an earlier GvL effect: although no guidelines can be drawn from the available studies, we addressed this strategy on page 17 (lines 636-653), discussed common practice, and added a specific article on this issue (Kim H.T. et al., Ref. 108)
  • With regard to the impact of ATG vs PTCy given as GVHD prophylaxis on the eventual outcome of HSCT: no direct comparison has been made where the issue was addressed independently from other important covariates, such as donor type, HSC source and type of conditioning; nevertheless we acknowledged the uncertainty of this issue on page 16 (line 595-600)

All changes in the Manuscript are highlighted in green.

Round 2

Reviewer 1 Report

Comments and Suggestions for Authors

Authors have satisfactorily addressed the issues raised